# Comparative Transcriptomics Identify Key Pituitary Circular RNAs That Participate in Sheep (*Ovis aries*) Reproduction

**DOI:** 10.3390/ani13172711

**Published:** 2023-08-25

**Authors:** Jianqi Yang, Jishun Tang, Xiaoyun He, Ran Di, Xiaosheng Zhang, Jinlong Zhang, Xiaofei Guo, Mingxing Chu, Wenping Hu

**Affiliations:** 1State Key Laboratory of Animal Biotech Breeding, Institute of Animal Science, Chinese Academy of Agricultural Sciences (CAAS), Beijing 100193, China; yangjianqi97@163.com (J.Y.); hedayun@sina.cn (X.H.); diran@caas.cn (R.D.); 2Institute of Animal Husbandry and Veterinary Medicine, Anhui Academy of Agricultural Sciences, Hefei 230031, China; tjs157@163.com; 3Tianjin Key Laboratory of Animal Molecular Breeding and Biotechnology, Tianjin Engineering Research Center of Animal Healthy Farming, Institute of Animal Science and Veterinary, Tianjin Academy of Agricultural Sciences, Tianjin 300381, China; zhangxs0221@126.com (X.Z.); jlzhang1010@163.com (J.Z.); guoxfnongda@163.com (X.G.)

**Keywords:** circular RNAs, pituitary, reproduction, sheep

## Abstract

**Simple Summary:**

Booroola fecundity gene (*FecB*) is known as the main gene for multiple lambs in sheep, which has been shown to affect the reproduction of Small Tail Han sheep. The three genotypes of *FecB* are BB (two copies of FecB mutation), +B (one copy of FecB mutation), and ++ (without FecB mutation). *FecB* ++ genotype Small Tail Han sheep usually have one offspring. Interestingly, we found that *FecB* ++ Small Tail Han sheep also had high-reproductive-capacity individuals (multiple lambs per litter). The pituitary affects reproductive processes of animals through a variety of hormones. How pituitary circular RNAs (circRNAs) affect fecundity in *FecB* ++ genotype sheep remains unknown. Through RNA sequencing, we speculated that changes in circRNAs may be related to the response of the pituitary to steroid hormones, and directly or indirectly affect the pituitary function and the fecundity of sheep. These results provide new insights into pituitary function and high fecundity in sheep.

**Abstract:**

CircRNAs have been found to play key roles in many biological processes and have diverse biological functions. There have been studies on circRNAs in sheep pituitary, and some important circRNAs have been found. But there are still few studies on circRNAs in sheep pituitary with different fecundity. In this study, we obtained the circRNAs expression profiles in the pituitary of *FecB* ++ genotype Small Tail Han sheep with different fecundity and estrous phases. A total of 34,878 circRNAs were identified in 12 pituitary samples, 300 differentially expressed circRNAs (DE circRNAs) (down: 104; up: 196) were identified in polytocous sheep in the follicular phase (PF) and monotocous sheep in the follicular phase (MF) (PF vs. MF), and 347 DE circRNAs (down: 162; up: 185) were identified in polytocous sheep in the luteal phase (PL) and monotocous sheep in the luteal phase (ML) (PL vs. ML). Cortisol synthesis and secretion pathway (follicular phase) and estrogen signaling pathway (luteal phase) were obtained by functional enrichment analysis of circRNAs source genes. Competing endogenous RNA (ceRNA) network analysis of key DE circRNAs revealed that oar-circ-0022776 (source gene *ITPR2*, follicular phase) targeted oar-miR-432, oar-circ-0009003 (source gene *ITPR1*, luteal phase) and oar-circ-0003113 (source gene *PLCB1*, luteal phase) targeted oar-miR-370-3p. We also explored the coding ability of DE circRNAs. In conclusion, our study shows that changes in the pituitary circRNAs may be related to the response of the pituitary to steroid hormones and regulate the reproductive process of sheep by affecting the pituitary function. Results of this study provide some new information for understanding the functions of circRNAs and the fecundity of *FecB* ++ genotype sheep.

## 1. Introduction

The litter size is an important economic character of sheep which directly determines the income of the breeder [1]. The *FecB* (fecundity Booroola, *BMPR1B*) gene is the major gene for multiple lamb trait of sheep obtained through research on Booroola sheep [2]. *FecB* mutation has an additive effect on enhancement of the ovine ovulation number and litter size, such that *FecB* ++ (without the *FecB* mutation) ewes ovulate one and birth one lamb, *FecB*B + (one copy of the *FecB* mutation) ewes may increase the number of ovulations by 1.5 and litter size by 1, while *FecB* BB (two copies of the *FecB* mutation) ewes will increase by 3 ovulations and 1.5 litter size [3]. But it is interesting that some *FecB* ++ Small Tail Han sheep also had high fecundity and maintained stable heredity in production [4].

Small Tail Han sheep is a native breed in China. They have the characteristics of multiple lambs per litter and year-round estrus, and the fecundity level of Small Tail Han sheep ewes is comparable to that of Booroola ewes. The origin of the *FecB* mutation in Small Tail Han sheep is different from Booroola ewes [5]. The study of *FecB* gene frequency showed that +B and BB genotypes were the dominant genotypes in this breed, and the frequency of B allele was greater than 0.5 [6]. *FecB* mutation has been shown to affect litter size in Small Tail Han sheep [7]. Existing studies have pointed out that the average litter size of Small Tail Han sheep primiparous ewes is 2.00, and the average litter size of pluriparous ewes is 2.61 [8]. Usually, *FecB* ++ ewes have only one offspring per litter, so it is an interesting finding that there are high-fertility individuals in *FecB* ++ populations. This may imply that there are other important genes and regulatory mechanisms that dominate this process.

The hypothalamic–pituitary–gonadal (HPG) axis finely regulates the reproductive process through the synthesis and release of multiple hormones [9]. Gonadotropin-releasing hormone (GnRH) secreted by the hypothalamus acts on the pituitary gland through the hypothalamus–pituitary portal system to stimulate the release of follicle-stimulating hormone (FSH) and luteinizing hormone (LH). FSH and LH serve as regulators of the ovary by driving folliculogenesis, estrogen (E2), and progesterone (P4) in the estrus cycle. These hormones also perform more complex functions through feedback regulatory mechanisms and hormone–hormone interactions [10]. As the center of the HPG axis, the pituitary communicates with the hypothalamus and ovary, and is one of the most important neuroendocrine organs. Some researchers have pointed out that at specific physiological times in the Booroola sheep breeding cycle, the plasma FSH concentration of high-producing ewes is higher than that of low-producing ewes [11,12,13]. In other words, some genes may influence the fecundity of ewes by affecting the amount and pattern of hormone secretion.

In addition to the coding region, there are also a lot of noncoding regions in the genome [14]. The ncRNAs (noncoding RNAs) like miRNAs (microRNAs), lncRNAs (long noncoding RNAs), and circRNAs are widely involved in various life activities [15,16,17]. CircRNAs has been extensively studied as an important ncRNA [18]. Most of them are formed by covalently closing the 3′ and 5′ ends of RNA, and are characterized by no polyA tail, insensitivity to exonucleases, and stronger stability than ordinary linear RNAs [19]. CircRNAs have multiple functions and usually act as a sponge for related miRNAs [20]. In the study on the oviduct of high-fecundity Yunshang black goats, it was found that circIQCG can affect the expression of *SMAD1* by sponging miR-145-5p, thereby promoting the activation of the TGF-β(transforming growth factor-β) pathway and affecting the fecundity of goats [21]. In addition, circRNAs can promote the transcription of source genes by interacting with U1 small ribonucleoproteins and RNA polymerase II [22]. CircRNAs can also bind RNA-binding proteins (RBPs) to perform their functions of translation, transcriptional regulation, and extracellular transport [23]. Some circRNAs have the potential to encode proteins, which breaks the cognition of circRNAs and they cannot be translated into proteins [24]. A recent study identified a large number of DE circRNAs between anestrous and estrous pituitary in sheep using RNA-seq. There are significant differences in the expression patterns of circRNAs in sheep pituitary during estrus and anestrus, suggesting that circRNAs may be closely related to the regulation of these two states [25]. However, little is known about the molecular mechanism of how circRNAs regulate pituitary function to affect *FecB* ++ Small Tail Han sheep fecundity.

Therefore, this study attempted to find out the key circRNAs involved in the reproductive process and reveal their potential mechanism of action by studying the expression profile of circRNAs in the pituitary of *FecB* ++ Small Tail Han sheep. This work is expected to provide new insights into the characteristics of high fecundity of Small Tail Han sheep without *FecB* mutation.

## 2. Materials and Methods

### 2.1. Ethics

In this research, all experiments were performed following the relevant regulations set by the CAAS (Chinese Academy of Agricultural Sciences, Beijing, China) and laws set by the People’s Republic of China. Ethical approval was provided by the Animal Ethics Committee of IAS-CAAS (No. IASCAAS-AE-03, 12 December 2016).

### 2.2. Animal Processing and Pituitary Collection

Based on TaqMan genotyping, 142 ewes with the *FecB* ++ genotype were identified from 890 Small Tail Han sheep [8]. Then, 12 ewes were selected from 142 *FecB* ++ genotype sheep, which were approximately 3 years old and similar in body features. All of them have a consecutive lambing record of 3 parities (Appendix A). The ovulation rate (the number of corpora lutea on both ovaries) was recorded for all 12 sheep in the four groups using laparoscopy on day 8 after CIDR removal in the previous estrous cycle when the formal experiment was conducted. This result confirmed the lambing records and ensured the accuracy of grouping [8]. These ewes were fed at a farm of the Tianjin Institute of Animal Sciences (117.2° E, 39.13° N) with free water and feed. Under natural temperature and light, this breed showed estrous from August to November and estrus synchronization was performed in September. Initially, 12 selected ewes were treated with CIDR (controlled internal drug releasing; Zoetis Australia Pty., Ltd., Sydney, NSW, Australia; progesterone 300 mg); in the meantime, vitamin AD was injected intramuscularly to protect the endometrium. After 12 days, the vaginal sponge was removed, and the removal time was set at 0 h. Based on previous lambing records and estrus cycle, the sheep were divided into two groups: six polytocous sheep and six monotocous sheep; three in each group were euthanized at 48 h (follicular phase) and the other three in each group were euthanized at 216 h (luteal phase). The timing of sampling was determined based on previous studies [8,26,27]. Pituitary samples were dissected immediately after euthanasia, frozen in liquid nitrogen, and stored at −80 °C for further analysis (Figure 1A).

### 2.3. RNA Isolation, Library Construction, and Sequencing

Total RNA was extracted from the pituitary samples of 12 ewes by using the TRIzol Reagent (Invitrogen, Carlsbad, CA, USA). Then, measurements of the RNA purity, concentration, and integrity were conducted by 1% agarose electrophoresis, a Kaiao K5500 spectrophotometer (Beijing Kaiao Technology Development Co., Ltd., Beijing, China). and RNA Nano 6000 Assay Kit of the Agilent Bioanalyzer 2100 System (Agilent Technologies, Santa Clara, CA, USA), respectively. The RIN (RNA integrity number) value of all the pituitary samples was greater than 7.

Three micrograms of total RNA was used as the starting amount for the construction of the library. Ribo-Zero™ Gold Kits (Epicenter, Madison, WI, USA) were used to remove rRNA (ribosomal RNA), and NEB Next Ultra Directional RNA Library Prep Kit for Illumina (NEB, Ispawich, MA, USA) was used to construct sequencing libraries for paired-end sequencing. Birefly, NEB Fragmentation Buffer was used to fragment RNA. The fragmented RNA will serve as a template for synthesizing the first strand of cDNA using random hexamers. Buffer, dNTPs, RNase H, and DNA Polymerase I were added to synthesize the second strand of cDNA. QiaQuick PCR-purified cDNA was end-repaired, labeled poly(A), and ligated into Illumina sequencing adapters. UNGase was used to digest the second strand of cDNA. After PCR amplification, the target fragment was recovered by agarose gel electrophoresis to obtain the final library. The libraries were loaded onto the Flowcell and placed in the sequencer. DNA fragments were amplified into millions of single-stranded DNA copies through the process of cluster generation. During sequencing by synthesis, sequences were identified by the binding of fluorescently labeled nucleotides to DNA template strands. After reading the forward DNA strand, the reads were washed away, and the process was repeated to read the reverse strand. This method is called paired-end sequencing. All the sequencing data were generated by Annoroad Gene Technology Co., Ltd. (Beijing, China) with Illumina HiSeq X platform (Illumina, San Diego, CA, USA).

### 2.4. Data Quality Control, Alignment, and CircRNA Identification

Raw reads were filtered to ensure the quality of further analytical data by using in-house Perl scripts (Annoroad Gene Technology Co., Ltd., Beijing, China). This process includes filtering out adaptor-polluted reads, low-quality reads, and reads with poly-N more than 5%. The clean data were attained after filtering, and statistics analyses were performed on their quantity and quality, including Q30 statistics, data quantity statistics, base content statistics, etc.

Firstly, the reference genome and annotation files (Oar v.3.1) were downloaded from ENSEMBL (http://www.ensembl.org/index.html, accessed on 10 April 2023). Then, reads were mapped to reference genome using BWA-MEM method by bwa (v0.7.9a) [28] with the parameter “mem -T 19 -t 4”. Subsequently, CIRI, an efficient and fast circRNAs identification tool [29], was used to identify circRNAs with default parameters.

### 2.5. Differential Expression Analysis

In general, SRPBM (spliced reads per billion mapping) was used to estimate the expression of circRNAs. To explore the differences between the groups, we used DESeq2 package (v.1.6.3) [30] to calculate log_2_ (fold change) and *p*-value based on the normalized counts. For screening key circRNAs, the thresholds of fold change > 1.5 and *p <* 0.05 were set to identify DE circRNAs. In summary, we counted the DE circRNAs between the different groups; these groups included polytocous sheep in the follicular phase (PF) vs. monotocous sheep in the follicular phase (MF) and polytocous sheep in the luteal phase (PL) vs. monotocous sheep in the luteal phase (ML).

### 2.6. Coding Potential Prediction of DE circRNAs

CircRNAs have the potential to encode protein. IRES (internal ribosome entry site) can recruit ribosomes, mediate the assembly of ribosome subunits, and, finally, realize circRNAs translation of polypeptides or proteins [31]. IRESfinder [32] is a Python package for identifying ribosome entry sites inside RNA in eukaryotic cells. This package was used to score the circRNAs sequence, and if it is greater than 0.5, it is considered to have a ribosome recruitment site required for encoding. In addition, multiple coding potential prediction tools were used to further identify the coding ability of circRNAs, including CPC2 [33], CNCI [34], CPAT [35], and PLEK [36]. The sequence of identified circRNAs is input in .fa file format; these software can identify whether the circRNAs are coding or noncoding.

### 2.7. Bioinformatics Analysis of Source Genes of DE circRNAs

Functional enrichment analysis was implemented to characterize source genes potential functions. This analysis process was realized through the online website where Gene Ontology (GO) enrichment was based on DAVID (https://david.ncifcrf.gov/, accessed on 12 May 2023) [37] and Kyoto Encyclopedia of Genes and Genomes (KEGG) pathway functional annotation was performed using KOBAS 3.0 (http://bioinfo.org/kobas/, accessed on 17 May 2023) [38]. A threshold of *p* < 0.05 was used as a criterion for the determination of whether the enrichment analysis was significant.

By screening the GO terms and KEGG pathways obtained through enrichment analysis, genes in GO terms and KEGG pathways related to reproduction or pituitary function will be selected. These genes and their encoded circRNAs will be potentially important.

### 2.8. Prediction of the Targeting miRNA of DE circRNAs

CircRNA can act as an miRNA sponge, which can reduce the inhibitory effect of miRNA on its target gene by adsorbing miRNA, and indirectly increase the expression level of the target gene. This regulatory network is called the ceRNA (competing endogenous RNA) mechanism. First, we obtained the mature sequences of all known miRNAs in sheep through the miRBase database (https://www.mirbase.org/, accessed on 25 May 2023) [39]. The interactions between DE circRNAs and all known miRNAs were predicted using miRanda (v3.3a) [40]. We adjusted the parameters of the software to “-sc 150” and “-en -15” to ensure the credibility of the prediction results. Finally, Cytoscape (v3.9.1) [41] was used to construct and visualize the targeting relationship between circRNAs and miRNAs.

### 2.9. Validation of DE circRNAs

Four DE circRNAs were randomly selected from each of the two comparison groups for real-time quantitative polymerase chain reaction (RT-qPCR) to verify the accuracy of the sequencing results. Primers were designed by using Primer Premier 6.0, synthesized by Sangon Biotech (Shanghai, China), and the internal reference gene was β-actin (Appendix A). First, 1 µg of total RNA from the samples was reverse-transcribed into cDNA using the PrimeScript™ RT reagent Kit (Takara, Beijing, China), and diluted five times. Real-time quantitative polymerase chain reaction (RT-qPCR) was performed by using the Roche Light Cycler^®^ 480 Ⅱ system (Roche Applied Science, Mannheim, Germany) following the instructions of TB Green^®^ Premix Ex Taq II (Takara, Beijing, China). RT-qPCR was performed under the following conditions: 95 °C for 5 min, followed by 40 cycles of amplification at 95 °C for 5 s and annealing at 60 °C for 30 s. Relative quantification of circRNA expression was compared to the internal reference gene and analyzed using the 2^−ΔΔCt^ method relying on a t-test. In addition, we also selected four differential components (oar-circ-0026172, oar-circ-0002134, oar-circ-0019011, and oar-circ-0021170) for looped detection to verify the results of RNA-seq. Briefly, the region containing the backsplice junction site of circRNA was amplified by reverse-transcription PCR (RT–PCR), and the RT–PCR product was subjected to Sanger sequencing [21].

## 3. Results

### 3.1. Summary Statistics of RNA-seq Data and Characteristic of circRNAs in Pituitary

We obtained a total of 1,451,839,480 raw reads after sequencing; the mapped reads numbered 1,451,057,999; the mapping rate of each sample reached more than 99.9% (Appendix A). We identified a total of 34,878 circRNAs from 12 pituitary samples (Appendix A). From the source of chromosomes, most of them came from chromosome 1, chromosome 2, and chromosome 3 (Figure 1B). These circRNAs came from a total of 5207 genes; 1573 genes encoded only one circRNA, accounting for 30% of the total gene count, and the number of genes encoded circRNAs less than or equal to 3 exceeded 50% of the total gene number (Figure 1C). From the perspective of the number of exons contained in circRNAs, circRNAs containing 2-6 exons all exceeded 1000 (Figure 1D). From the perspective of type, about 70% of circRNAs were CLASSIC (the formation sites of circRNAs are all on the boundaries of exons) (Figure 1E).

### 3.2. Identification of DE circRNAs

To identify differentially expressed circRNAs, we set thresholds of fold change > 1.5 and *p <* 0.05 as criteria. The 300 DE circRNAs was identified in PF vs. MF; 196 were upregulated and 104 were downregulated (Figure 2A, Appendix A). The 347 DE circRNAs were identified; 185 were upregulated and 162 were downregulated in PL vs. ML (Figure 2B, Appendix A). We present this result using visualizations showing differences in expression patterns between PF vs. MF (Figure 2C) and between PL vs. ML (Figure 2D). In addition, the RT-qPCR results of DE circRNAs showed a similar expression trend to the sequencing results, which proved the accuracy of our sequencing results (Figure 3). The results of RT-PCR showed that the sequence information of the junctions were same as RNA-seq, which proved that circRNAs were looped (Appendix A).

### 3.3. Identifying the Coding Potential of DE circRNAs

The IRESfinder software firstly judged and scored the ribosome recruitment ability of DE circRNAs sequences. In the PF vs. MF, a total of 147 differential circRNAs were predicted to contain IRES with a score greater than 0.5, and in the PL vs. ML group, there were 165. Subsequently, the coding potential prediction software CPC2, CNCI, CPAT, and PLEK were used to analyze and summarize the coding ability of DE circRNAs. CPC2, CNCI, and PLEK give the results of DE circRNAs as yes or no in the form of tags, and CPAT gives the results in the form of scores. In summary, we obtained the final result under the common conditions that IRES is included and the score is greater than 0.5, the labels of CPC2, CNCI, and PLEK are coding, and the CPAT score is greater than 0.5 (Appendix A). A total of 34 circRNAs had potential coding ability in the PF vs. MF (Figure 4A), while there were 31 in the PL vs. ML (Figure 4B).

### 3.4. Functional Enrichment for Source Genes of DE circRNAs

Through GO and KEGG enrichment analysis of source genes of DE circRNAs, a deeper understanding of gene function can be achieved. GO enrichment analysis mainly refers to the results obtained from the online website DAVID (Appendix A). A total of 47 items were obtained from the GO analysis of source genes of DE circRNAs in PF vs. MF, including 16 biological process (BP) items, 12 cell components (CC) items, and 19 molecular functions (MF) items, among which *p*-value < 0.05 is significant for a total of 28 items (Figure 5A). It is worth noting that this includes a star pathway TGF-β (GO:0007179) related to sheep high yield, and four genes, *ITGB8*, *USP15*, *USP9X,* and *ZFYVE9*, were enriched in this GO entry. A total of 48 items were obtained from the GO enrichment analysis of source genes of DE circRNAs in PL vs. ML, including 14 BP items, 17 CC items, and 17 MF items, among which *p* < 0.05 was significant for a total of 31 items (Figure 5B).

KEGG enrichment analysis mainly refers to the results obtained from the online website KOBAS 3.0 (Appendix A). A total of 164 items were obtained in the KEGG analysis of source genes of DE circRNAs in PF vs. MF, of which 7 items were significant according to *p* < 0.05 (Figure 6A). Among them, cortisol synthesis and secretion (oas04927) were directly related to the function of the pituitary gland. Genes enriched in this pathway include *ITPR2*, *PDE8B*, and *PLCB1*. According to the description of the pathway on the KEGG website, this pathway was also involved in the response of the pituitary gland to steroids. A total of 191 items were obtained in the KEGG analysis of source genes of DE circRNAs in PL vs. ML, of which 14 items were significant according to *p* < 0.05 (Figure 6B). Among them, the glucagon signaling pathway (oas04922) and estrogen signaling pathway (oas04915) are directly related to pituitary function. Genes enriched in these two pathways included *AKT3*, *PDE3B*, *ITPR1*, *PLCB1*, *ATF2*, and *ESR1*. In addition, in KEGG enrichment analysis of the two groups, other items related to pituitary function and reproduction were also obtained, such as the GnRH signaling pathway (oas04912), Oocyte meiosis (oas04114), etc. But these items were not significant (*p* > 0.05).

It is worth noting that although these pathways were not significant, some genes involved in these pathways were shared with significant pathways, namely, those genes just mentioned. Therefore, the changes in these genes, as well as in the circRNAs they encode, might have a greater impact on reproductive performance. Both DAVID and KOBAS can perform GO and KEGG analysis at the same time to annotate the function of genes, but DAVID has fewer KEGG items, while KOBAS has fewer GO items. We synthesized the two descriptions of gene functions and selected all items directly related to pituitary function and reproduction for integration and visualization, such as some hormone-release items (Appendix A). *ITPR2*, *SHC4*, *PLCB1,* and *GSK3B* appeared in multiple pathways in PF vs. MF (Figure 7A). There were two circRNAs derived from the USP15 gene (oar-circ-0022268, oar-circ-0022266), so the log_2_ (fold change) is not shown in the figure. *ESR1*, *PDE3B*, *ATF2*, *ITPR1*, *AKT3*, and *PLCB1* were present in multiple pathways in PL vs. ML (Figure 7B). There are two circRNAs derived from the *ATK3* gene (oar-circ-0002271, oar-circ-0002289), and the log_2_ (fold change) is not shown in the figure. Interestingly, these two circRNAs have the opposite trend of change.

To sum up, those genes that appeared in enriched KEGG pathways or GO items and at the same time appeared frequently in KEGG pathways or GO items directly related to pituitary function would be the targets we focus on. In PF vs. MF, *ITPR2* (encoded oar-circ-0022776) and *PLCB1* (encoded oar-circ-0003112) were the focus of our attention. In PL vs. ML, *ESR1* (encoding oar-circ-0032298), *PDE3B* (encoding oar-circ-0005399), *ATF2* (encoding oar-circ-0018201), *ITPR1* (encoding oar-circ-0009003), *AKT3* (encoding oar-circ-0002271 and oar-circ-0002289), and *PLCB1* (code oar-circ-0003113) were the focus of our attention.

### 3.5. CircRNA–miRNA Coexpression Network

We used miRanda software to predict the binding abilities between the mature sequences of all known miRNAs in sheep in the miRBase database and the obtained DE circRNAs. The miRanda software mainly judges the binding ability of the two by evaluating the degree of sequence complementary matching and the free energy of the formed composite structure. We adjusted the parameters: “-sc” and “-en” obtained different results; to ensure a higher confidence, the main reference parameter is the result of “-sc 150 -en -15” (Appendix A). A total of 718 circRNA–miRNA matching relationship pairs were obtained from the DE circRNAs in the follicular phase, and the most frequently occurring miRNAs in these relationship pairs were oar-miR-432 and oar-miR-134-3p, both 20 times (Figure 8). A total of 852 circRNA–miRNA matching relationship pairs were obtained for luteal phase differential circRNAs, and the miRNAs that appeared more in these relationship pairs were oar-miR-329b-5p and oar-miR-370-3p, both more than 30 times (Figure 9).

We selected the circRNAs encoded by the genes of interest and their relationship pairs, that is, the genes that were finally screened in the functional enrichment analysis, and visualized the gene–circRNA–miRNA relationship using Cytoscape (v3.9.1). Only *ITPR2* (encoding oar-circ-0022776) was predicted in 8 circRNA–miRNA relationship pairs in the follicular phase (Figure 10A), and a total of 23 pairs were predicted in the luteal phase (Figure 10B). *ESR1* (encoding oar-circ-0032298) and *AKT3* (encoding oar-circ-0002271) were not predicted in the result. *AKT3* encoded two different circRNAs that have opposite changing trends and different miRNA-binding abilities. *PLCB1* encoded different circRNAs in the two periods (follicular phase: oar-circ-0003112, luteal phase: oar-circ-0003113) and had different miRNA-binding capabilities; only oar-circ-0003113 had predicted results.

## 4. Discussion

### 4.1. Characteristics of Pituitary circRNAs

The pituitary communicates with the hypothalamus and ovary, participates in the reproductive process in the form of hormone secretion, and also receives hormone signals from the hypothalamus and ovary, forming a complex regulatory relationship [42]. In previous studies, researchers analyzed the transcriptomes of estrus and nonestrus sheep pituitary to explore the regulatory mechanism of circRNA in sheep pituitary [25]. Compared with previous studies that focused on different physiological stages, we focused more on the phenotype of fecundity, and hoped to obtain important circRNAs in sheep pituitary under the control of genetic background (no FecB mutation). From the perspective of circRNAs characteristics, the total number of circRNAs we obtained was more than that of previous studies. But compared to our previous studies of circRNAs in different tissue samples (hypothalamus and uterus) of the same individual [43,44], the total number of circRNAs identified was similar. This may include a variety of reasons, such as differences in samples and differences in analysis procedures, which also need to be considered in future research.

Similar numbers of differential circRNAs were obtained from the polytocous and monotocous comparison groups at different periods, but the proportions of upregulated and downregulated DE circRNAs to the total were different. A total of 300 DE circRNAs were obtained in the follicular phase comparison group, of which 196 were upregulated, accounting for about two-thirds of the total, and 104 were downregulated, accounting for about one-third of the total. A total of 347 DE circRNAs were identified in the luteal phase comparison group, of which 185 were upregulated and 162 were downregulated, and the difference in the number of upregulated and downregulated DE circRNAs accounted for about half each. This also shows that circRNAs have different functions and regulatory mechanisms between different periods, which is similar to previous research results [17,25].

### 4.2. Functional Enrichment Analysis of DE circRNAs of Their Source Genes

Previous studies have shown that the main genes that control high fecundity in sheep include *BMPR1B*, *BMP15*, *GDF9*, etc., and they function through BMP/Smad and TGF-β/Smad pathways [45]. But there are also high levels in sheep that do not contain FecB mutation, which is also the point of interest for us to carry out this study. We performed functional enrichment analysis on the source genes of DE circRNAs and obtained some interesting results. The GO enrichment analysis of DE circRNAs-derived genes in the follicular phase showed that some genes (*ITGB8*, *USP15*, *USP9X,* and *ZFYVE9*) of the TGF-β pathway (GO:0007179) affected the high fecundity of sheep. That is to say, we cannot completely avoid genes or related pathways that are known to have an impact on high fecundity of sheep, but it is still necessary and exciting to discover new genes or pathways.

Just like previous studies on the function of the pituitary, we initially expected that the reason for the high fecundity of sheep might be related to the secretion of FSH and LH hormones [8,46]. The KEGG analysis of differentially expressed DE circRNAs source genes in PF vs. MF was enriched to the cortisol synthesis and secretion (oas04927), which is also involved in the response of the pituitary to steroid and participate in the breeding process of sheep. In the breeding season and nonbreeding season, estradiol can make cortisol act directly on the pituitary, inhibiting the response of sheep pituitary gland to GnRH [47]. The KEGG analysis of DE circRNA source genes in PL vs. ML was enriched to the estrogen signaling pathway (oas04915). Estrogen is one of the most important hormones in animal reproduction, affecting animal estrus, oocyte development and ovulation, and other links [48]. In summary, our results show that the pituitary may affect the secretion of FSH and LH hormones through the response to estrogen and may finally affect the fertility of the experimental population in this study.

### 4.3. Functional Exploration of Key circRNAs and Their Source Genes

The functional exploration of circRNAs from important gene sources has been carried out from multiple perspectives. CircRNAs are known to promote the transcription of source genes by interacting with U1 small ribonucleoproteins (snRNPs) and RNA polymerase II [22]. Therefore, identification of the potential functions of source genes associated with circRNAs may shed light on their functions.

According to the results of enrichment analysis, we selected some key genes and their encoded circRNAs. *ITPR1* and *ITPR2* both belong to the *ITPR* family and were enriched in two periods, respectively. This gene family is directly related to the calcium ion signal transduction of cells [49]. *ITPR1* has been shown to play an important role in various biological reproductive processes such as the number of eggs laid by female broilers [50], the maintenance of testicular function in male mice [51], the quality of rabbit oocyte [52], and capacitation of goat sperm [53]. In the study of the ovarian function of the blunt snout bream Megalobrama amblycephala [54], the biosynthesis of steroid hormones in the ovaries of adult individuals was increased compared with juvenile individuals, and the genes involved in the “GnRH signaling pathway” in the corresponding mature females included *ITPR2*, *CAMK2,* and *MEKK1,* which were significantly upregulated. This result supports our hypothesis that the pituitary responds to estrogen to affect fecundity. *PLCB1* encoded two circRNAs, which were different in the high- and low-yield comparison groups in the two periods. The *PLCB1* gene may be associated with litter size in sheep in a genome-wide association study (GWAS) of six sheep breeds [55]. In another study on sheep seasonal estrus, by constructing ovariectomy model and exogenous estrogen treatment, it was also shown that *PLCB1* regulates sheep hypothalamus function and reproductive process by responding to estrogen and affecting hormone secretion during sheep seasonal reproduction [56]. Considering that the reproductive process is coregulated by changes in time and space of multiple hormones, the importance of pathways related to cell signaling and communication seems logical, and may explain why these genes, which play a role in signal transduction, were found.

CircRNAs can act as miRNA sponges and function in the form of ceRNA networks. Oar-miR-432 was the most targeted miRNA by DE circRNAs in the follicular phase. For our selected circRNAs, oar-circ-0022776 (source gene *ITPR2*) also targets oar-miR-432. In the study of ovary circRNAs in sheep with different fecundity, oar-miR-432 was predicted to be the targeting miRNA of DE circRNAs [57]. Oar-miR-432 was also screened in our previous study on the hypothalamus of sheep with different fecundity [58]. It was predicted that oar-miR-432 may be related to the release of GnRH hormone and thus affect the reproductive performance of sheep. The high-frequency oar-miR-370-3p in the luteal phase was located in the study of DE miRNA in the adrenal glands of high- and low-yield sheep, showing that it was related to sheep fecundity [59]. Oar-circ-0018201 (source gene *ATF2*), oar-circ-0009003 (source gene *ITPR1*), and oar-circ-0003113 (source gene *PLCB1*) could target oar-miR-370-3p. Oar-miR-16b targeted by oar-circ-0018201 (source gene *ATF2*) has been shown to regulate the release of FSH and LH from the sheep pituitary by affecting *SMAD2* and TGF-β/SMAD2 pathways [60]. These results can support our conjecture, but further research and confirmation are still needed.

As a novel mode of action, we explored and predicted the coding ability of DE circRNAs. The circRNAs we focused on have no coding ability, but this does not prevent other differential circRNAs from functioning in the form of coding proteins.

### 4.4. The Clinical Significance of the Findings

Our study of Small Tail Han sheep pituitary circRNAs enriches the understanding of pituitary function. It also provides a preliminary exploration to understand how pituitary circRNAs affects the reproductive ability of Small Tail Han sheep pituitary. Through GO, KEGG, and ceRNA regulatory network analysis methods, some key circRNAs were located in the pituitary circRNAs expression profile. These circRNAs can be used as further research objects, which may be a breakthrough in understanding why some *FecB* ++ Small Tail Han sheep individuals still have high fecundity.

## 5. Conclusions

We obtained the circRNA expression profiles of the pituitary glands of *FecB* ++ Small Tail Han sheep at different fertility and estrous cycle. Through the enrichment analysis of DE circRNA source genes, ceRNA network analysis explored the functions of key DE circRNAs. These results indicate that at the level of circRNA, the pituitary may affect the secretion of FSH and LH hormones through the response to estrogen, and finally affect the fecundity of sheep. This study provides new insights into the mechanism by which the pituitary regulates sheep reproduction, especially in high-yielding sheep without *FecB* mutation, but these results still need further validation and research.

## Figures and Tables

**Figure 1 animals-13-02711-f001:**
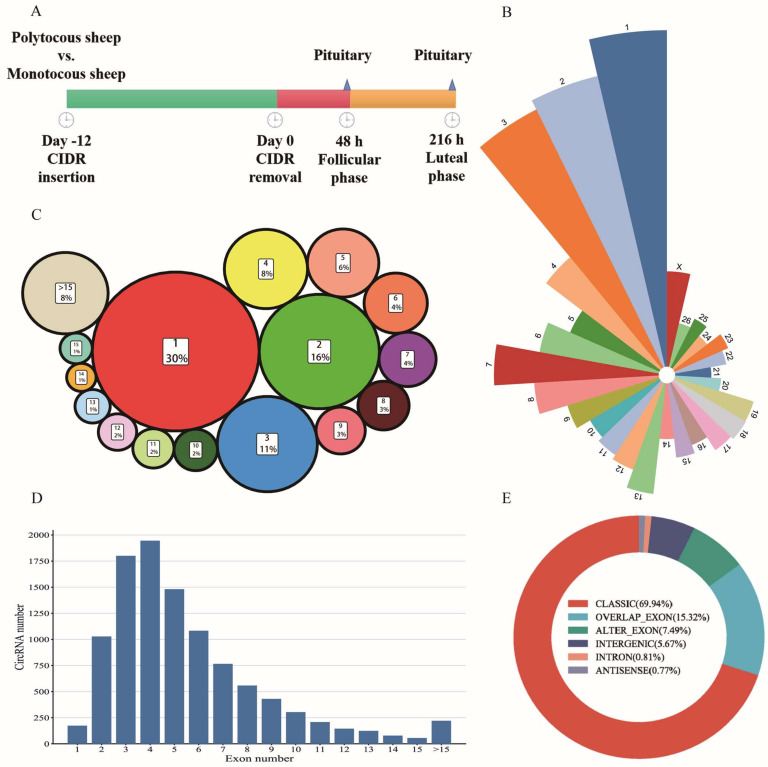
Characteristic of circRNAs in the pituitary. (**A**) Schematic representation of pituitary samples collected designed in this study. (**B**) Quantity statistics of circRNAs from different chromosome sources. Different colors represent different chromosomes, and the area represents the number of circRNAs. (**C**) Statistics of the number of genes encoding different numbers of circRNAs. The number is the count of encoded genes, and the area is the proportion of the total encoded genes. (**D**) Exon number characteristics of circRNAs. (**E**) Type characteristics of circRNAs.

**Figure 2 animals-13-02711-f002:**
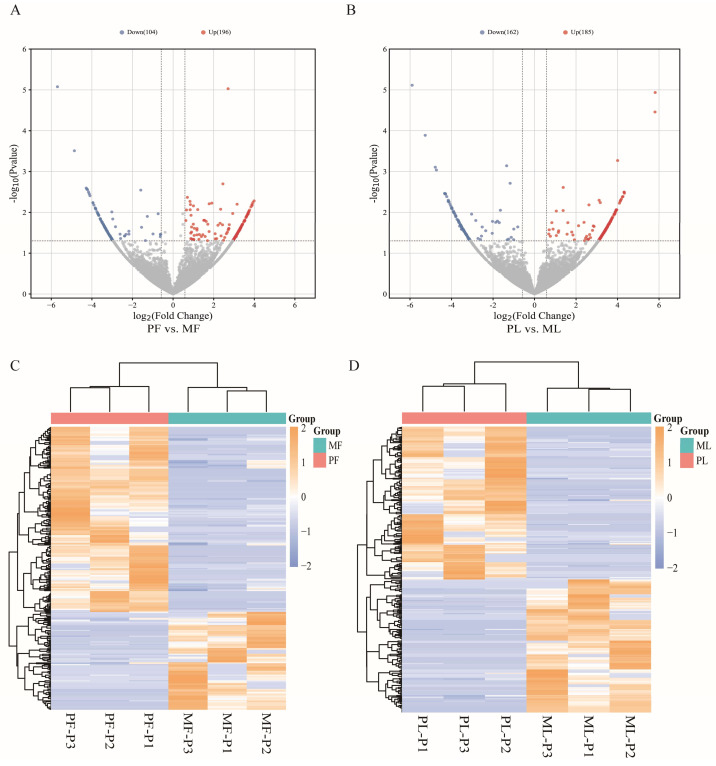
Differential expression analysis of circRNAs. (**A**,**B**) Volcano plots showing the upregulated and downregulated circRNAs in PF vs. MF and PL vs. ML. (**C**,**D**) The expression pattern of DE circRNAs and hierarchical clustering analysis in PF vs. MF and PL vs. ML.

**Figure 3 animals-13-02711-f003:**
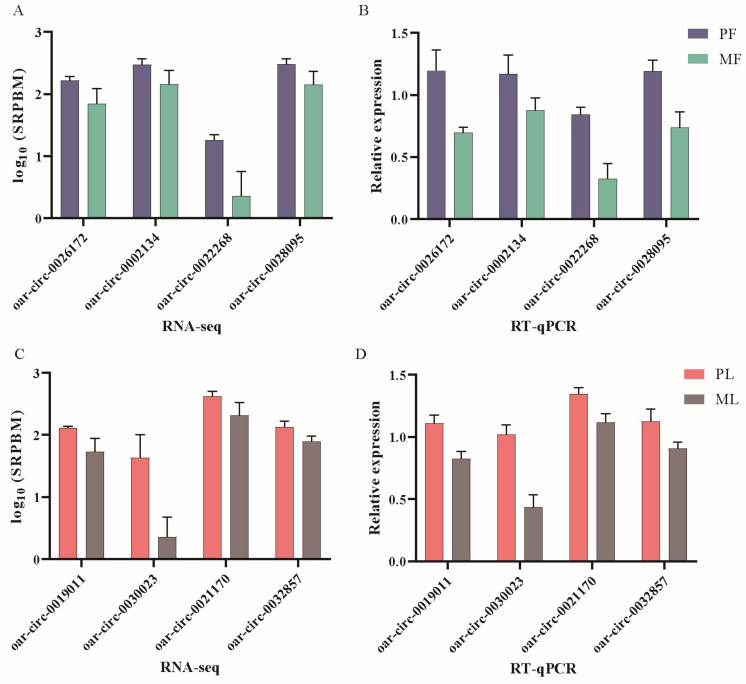
Validation of RNA-sequencing (RNA-seq) data using reverse-transcription real-time quantitative polymerase chain reaction (RT-qPCR). (**A**,**C**) RNA-seq results of eight selected circRNAs in PF vs. MF and PL vs. ML. (**B**,**D**) RT-qPCR results of eight selected circRNAs in PF vs. MF and PL vs. ML.

**Figure 4 animals-13-02711-f004:**
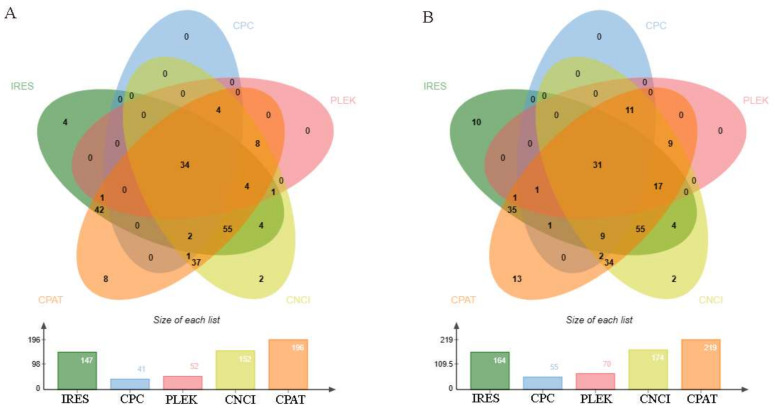
Coding potential prediction of DE circRNAs in PF vs. MF (**A**) and in PL vs. ML (**B**). The numbers in the figure represent the number of circRNAs that were predicted to have potential coding capabilities.

**Figure 5 animals-13-02711-f005:**
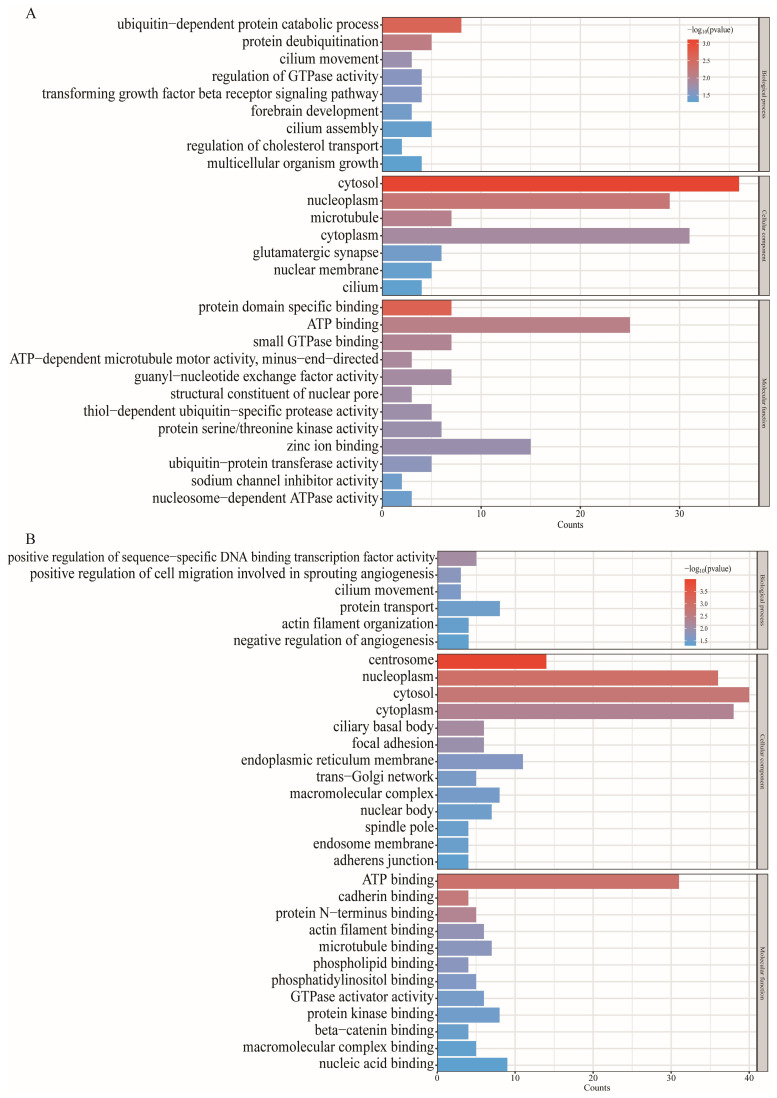
Gene Ontology functional enrichment analysis. (**A**) The enriched Gene Ontology (GO) terms in PF vs. MF. (**B**) The enriched Gene Ontology (GO) terms in PL vs. ML.

**Figure 6 animals-13-02711-f006:**
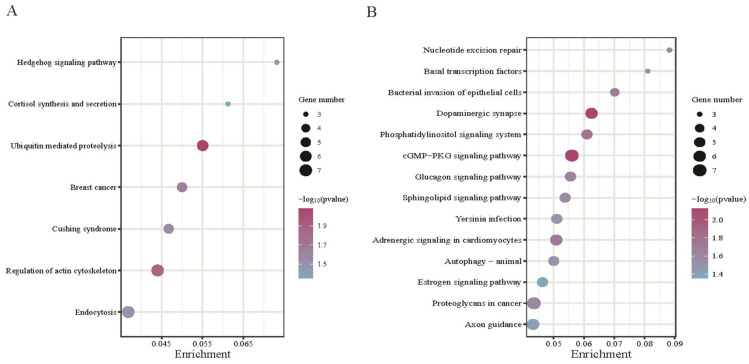
Kyoto Encyclopedia of Genes and Genomes (KEGG) enrichment analysis. (**A**) The enriched KEGG pathways in PF vs. MF. (**B**) The enriched KEGG pathways in PL vs. ML.

**Figure 7 animals-13-02711-f007:**
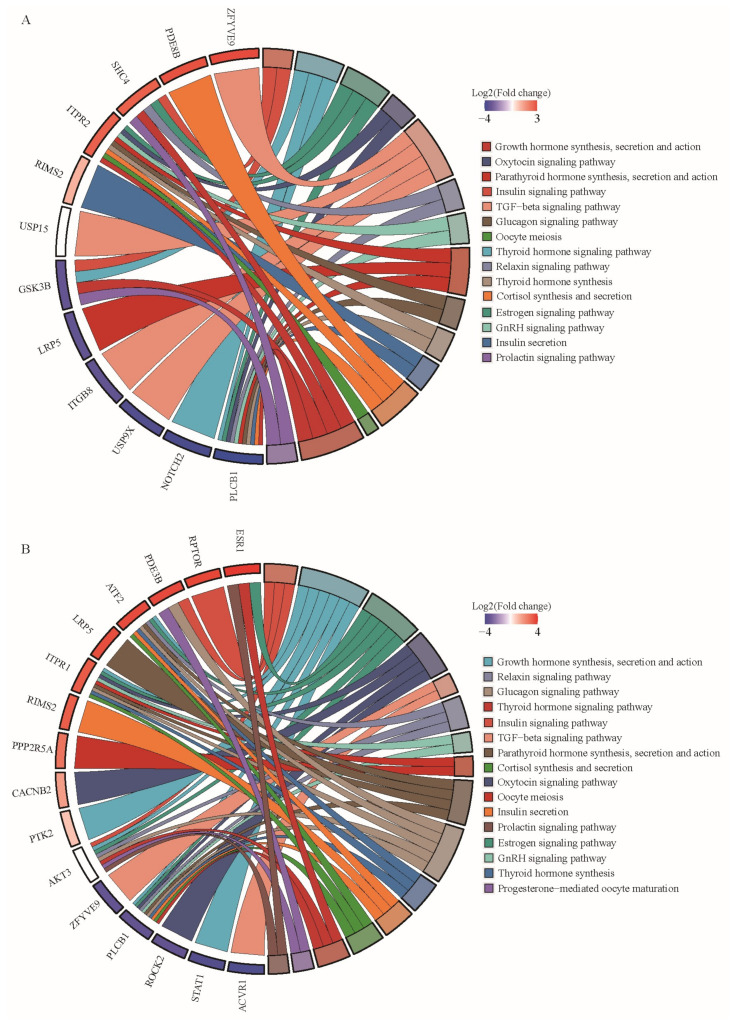
Screening of important DE circRNAs and their source genes. (**A**) Pathways directly related to pituitary function and genes in these pathways in PF vs. MF. (**B**) Pathways directly related to pituitary function and genes in these pathways in PL vs. ML.

**Figure 8 animals-13-02711-f008:**
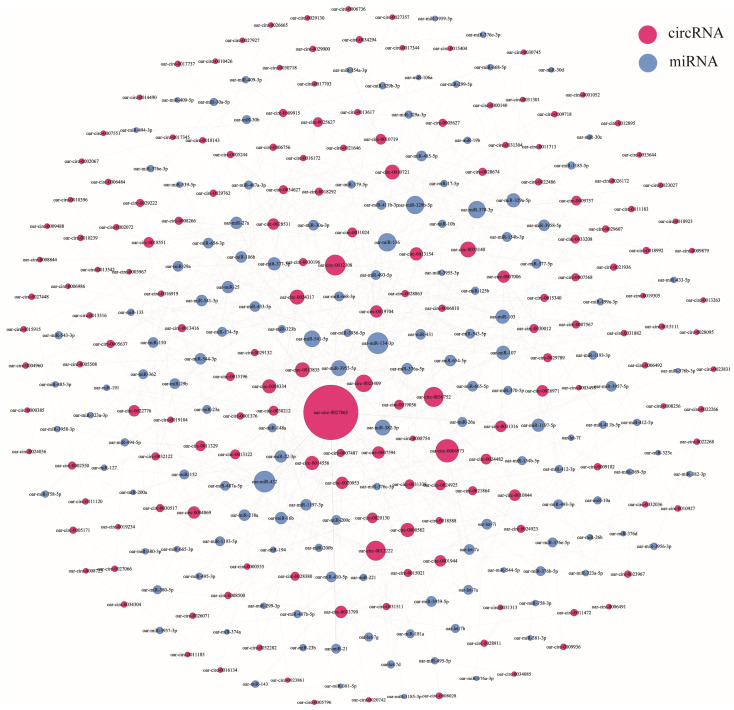
Relationship between DE circRNAs and their targeted miRNAs in the follicular phase.

**Figure 9 animals-13-02711-f009:**
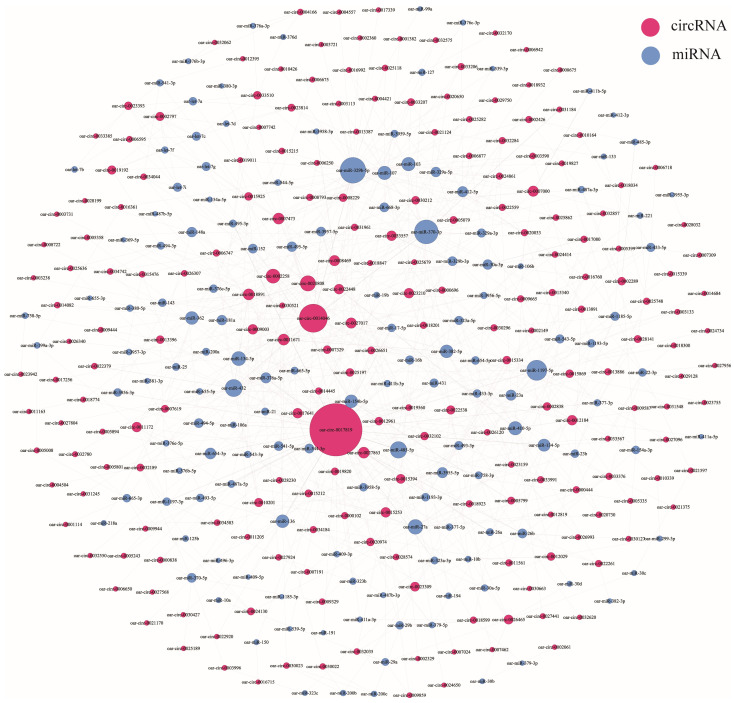
Relationship between DE circRNAs and their targeted miRNAs in the luteal phase.

**Figure 10 animals-13-02711-f010:**
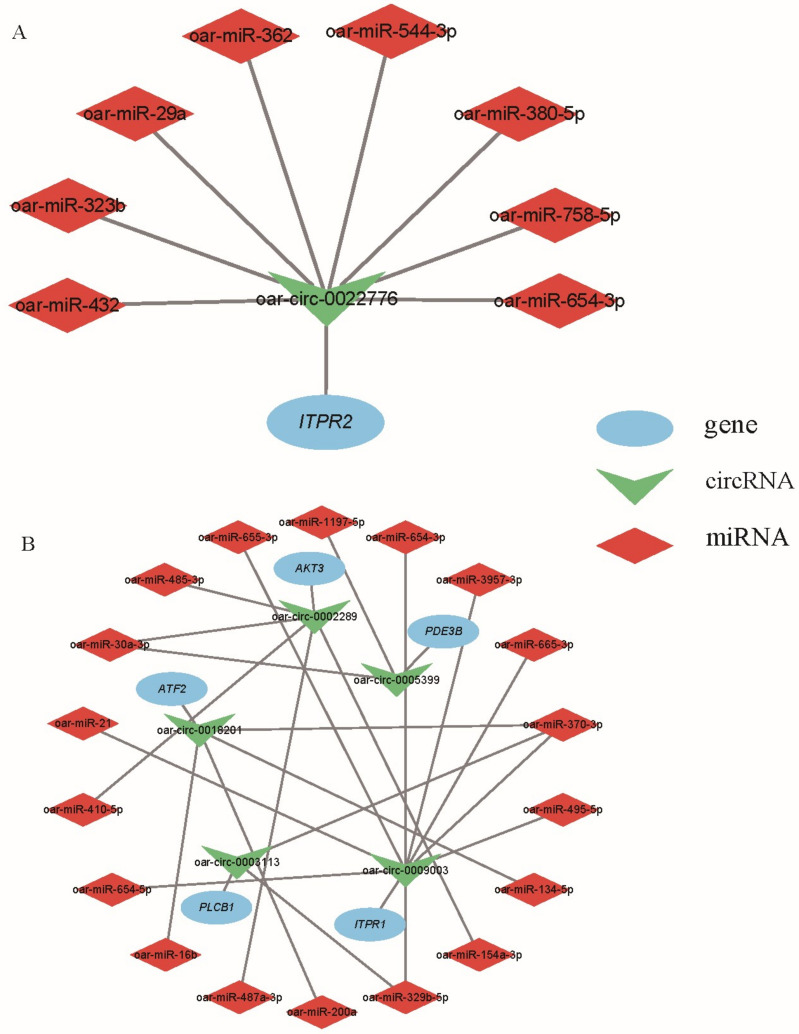
Regulatory network consisting of selected important circRNAs, their source genes, and their targeted miRNAs. (**A**) Relationship pairs in PF vs. MF. (**B**) Relationship pairs in PL vs. ML.

## Data Availability

All the data obtained from RNA-seq have been deposited in the Sequence Read Archive database under the bioproject numbers PRJNA993367.

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
