# Peer review of "Comparative Transcriptomics Identify Key Pituitary Circular RNAs That Participate in Sheep (Ovis aries) Reproduction"

_animals, 2023, doi:10.3390/ani13172711_

Round 1

Reviewer 1 Report

This is an interesting manuscript aimed to compare circRNAs expression profiles in sheep pituitary with different fecundity. Methodology and results are clear, and discussion is well described and supported. I suggest clarifying minor details in Methods and Conclusions sections. Also suggest considering next minor comments:

-       Lines 36-37: Abstract usually should not include results from other studies.

-       Line 40: Replace “These results of this study will provide” by “Results of this study provided”.

-       Line 50: Remove the word “And”.

-       Line 54: Replace “will increase by 3 and 1.5” by “will increase by 3 ovulations and 1.5 litter size”.

-       Line 67: Replace “There were” by “Some”.

-       Line 70: Replace “affect” by “influence”.

-       Line 110: Replace “is” by “show”.

-       Line 132: How the sequencing process was performed?, please briefly explain each step.

-       Line 151: Did the authors include a test to adjust p-values of circRNAs for false discovery rate?

-       Line 389: Replace “sand” by “and”.

-       Line 414: Replace “significantly up-regulation” by “which were significantly up-regulated”.

-       Line 444: I suggest removing results from Conclusions section as they were previously mentioned (i.e., Results section).

Author Response

Dear Reviewer:

Thank you for your comments concerning our manuscript entitled“Comparative Transcriptomics Identify Key Pituitary Circular RNAs that Participate in Sheep (Ovis aries) Reproduction“ (ID: animals-2528678). These comments are very valuable and helpful for revising and improving our paper, as well as providing important guidance to our research work. We have studied the comments carefully and have made corrections which we hope to meet with approval. Revised portions are marked in red in the manuscript. The main corrections in the paper and the response to your comments are as flowing:

Point 1: Lines 36-37: Abstract usually should not include results from other studies.

Response: We are sorry for our mistakes in the manuscript. This content has been removed(Lines 49-50).

Point 2: Line 40: Replace “These results of this study will provide” by “Results of this study provided”.

Response: Thanks for your comments. This content has been modified(Line 53).

Point 3: Line 50: Remove the word “And”.

Response: Thank you for your suggestion. This content has been removed(Line 64).

Point 4: Line 54: Replace “will increase by 3 and 1.5“ by “will increase by 3 ovulations and 1.5 litter size“.

Response: Thank you for your careful work. This content has been modified(Line 68).

Point 5: Line 67: Replace “There were“ by “Some“.

Response: Thanks for your comments. This content has been modified(Line 93).

Point 6:  Line 70: Replace “affect“ by “influence“.

Response: Thank you for your suggestion. This content has been modified(Line 96).

Point 7:  Line 110: Replace “is“ by “show“.

Response: Thank you for your comments. This content has been modified(Line 140).

Point 8: Line 132: How the sequencing process was performed?, please briefly explain each step.

Response: Thank you for your valuable comments. We supplemented the sequencing process, including library construction and on-machine sequencing(Lines 163-175). We used the Illumina HiSeq X platform, so the supplementary steps refer to the official introduction of Illumina, which ensured the accuracy of the content(https://www.illumina.com.cn/).

Point 9: Line 151: Did the authors include a test to adjust p-values of circRNAs for false discovery rate?

Response: Thank you for your careful work. We provided the adjust p-values in the supplementary material(Table S5 DE circRNAs identified in pituitary tissues). In addition to Fold Change, adjust p-values will also be selected to screen the difference components. But different studies will choose different threshold ranges. Generally speaking, the determination of the threshold is for the research content, and the number of difference components can be determined by different thresholds. Our study also included a test to adjust p-values of circRNAs for false discovery rate. However, the results were poor, and differential circRNAs cannot be effectively selected. So, the thresholds of fold change > 1.5 and P < 0.05 were set to identify DE circRNAs(Line 194).

Point 10:  Line 389: Replace “sand“ by “and“.

Response: Thanks for your comments. This content has been modified(Line 438).

Point 11: Line 414: Replace “significantly up-regulation“ by “which were significantly up-regulated“.

Response: Thanks for your comments. This content has been modified(Line 463).

Point 12:  Line 444: I suggest removing results from Conclusions section as they were previously mentioned (i.e., Results section).

Response: Thank you for your valuable and thoughtful comments. Based on your suggestion, we have deleted the results in the conclusion section and rewritten the conclusion section.(Line 502).

Thank you for your valuable comments. We have made every effort to enhance the manuscript and incorporated some revisions accordingly while ensuring that the content and structure of the paper remain unchanged. We sincerely appreciate the diligent work of yours and hope that our amendments will meet with your approval. Once again, thank you very much for your insightful suggestions.

Reviewer 2 Report

The authors identified the differential expressed circular RNAs in pituitary from sheep with different fecundity. The job is interesting. I only have several comments here.

1. Could you please add some considerations for you experimental design? Why you choose 48h and 216h? What is the criteria on sample group for polytocous and monotocous? And is this criteria only based on previous lambing revoreds and estrus cycle? Did this have any other methods?

2. Have you determined the differential-expressed RNAs you identified were circular RNA?

3. Why not do more research on these differential-expressed RNAs? More wet-lab would be better.

Author Response

Dear Reviewer:

Thank you for your comments concerning our manuscript entitled“Comparative Transcriptomics Identify Key Pituitary Circular RNAs that Participate in Sheep (Ovis aries) Reproduction“ (ID: animals-2528678). These comments are very valuable and helpful for revising and improving our paper, as well as providing important guidance to our research work. We have studied the comments carefully and have made corrections which we hope to meet with approval. Revised portions are marked in red in the manuscript. The main corrections in the paper and the response to your comments are as flowing:

Point 1: Could you please add some considerations for you experimental design? Why you choose 48h and 216h? What is the criteria on sample group for polytocous and monotocous? And is this criteria only based on previous lambing revoreds and estrus cycle? Did this have any other methods?

Response: Thank you for your careful work. We chose 48 hours and 216 hours as the sampling time points for the following reasons. First of all, if you want to determine the follicular phase and luteal phase, you must first determine the time of ovulation. In the previous study of our team, the ovulation time of different genotypes of Small-tailed Han sheep was studied, and the interval from the removal of CIDR to the first ovulation was about 52-63 hours(Wang X, Guo X, He X, et al. Effects of FecB Mutation on Estrus, Ovulation, and Endocrine Characteristics in Small Tail Han Sheep. Front Vet Sci. 2021;8:709737. Published 2021 Nov 22. doi:10.3389/fvets.2021.709737). So the follicular phase sampling time should be before this. Secondly, in previous studies on the estrous cycle of ewes(Bartlewski PM, Baby TE, Giffin JL. Reproductive cycles in sheep. Anim Reprod Sci. 2011;124(3-4):259-268. doi:10.1016/j.anireprosci.2011.02.024), it was pointed out that the corpus luteum reached its maximum diameter 6 days after ovulation, and shrank about 12-15 days after ovulation, which provided a reference for the selection of sampling time points during the luteal phase. In similar studies, other investigators have chosen similar sampling times(Wan Z, Yang H, Cai Y, et al. Comparative Transcriptomic Analysis of Hu Sheep Pituitary Gland Prolificacy at the Follicular and Luteal Phases. Genes (Basel). 2022;13(3):440. Published 2022 Feb 27. doi:10.3390/genes13030440), justifying this experimental design. In summary, we determined 48 hours and 216 hours as the sampling time points.

In addition to previous lambing and estrus cycles, we also referred to laparoscopic observations. This approach was borrowed from our previous research(Wang X, Guo X, He X, et al. Effects of FecB Mutation on Estrus, Ovulation, and Endocrine Characteristics in Small Tail Han Sheep. Front Vet Sci. 2021;8:709737. Published 2021 Nov 22. doi:10.3389/fvets.2021.709737). The ovulation rate (the number of corpora lutea on both ovaries) was recorded for all 12 sheep in the four groups using laparoscopy on day 8 after CIDR removal in the previous estrous cycle when the formal experiment was conducted. This result confirmed the lambing records and ensured the accuracy of grouping.

In order to avoid confusion for readers, we have added a description of the source of the method and references to the literature in the article(Lines 134-138, 148).

Point 2: Have you determined the differential-expressed RNAs you identified were circular RNA?

Response: Thank you for your valuable and thoughtful comments. CircRNA is a new type of RNA molecule that is different from linear RNA. Therefore, the key point to identify circRNA is to identify the backsplice junction site.

Firstly, the CIRI software was used for identification. This process used the BWA-MEM algorithm for sequence splitting and aligning, and then scanned the SAM files of the alignment results to find PCC (paired chiastic clipping), PEM (paired-end mapping) sites, and GT-AG splicing signals. The sequences with the junction site were re-aligned with the dynamic programming algorithm to ensure the reliability of identifying circRNA.

Secondly, in order to ensure the accuracy of the RNA-seq, we randomly selected four circRNAs (oar-circ-0026172, oar-circ-0002134, oar-circ-0019011, oar-circ-0021170), designed specific RT-PCR primers for the junction region, and performed Sanger sequencing on the RT-RCR products. The results showed that the sequence information of the junctions were same as RNA-seq, which proved that circRNAs were looped.

These methods referred to previous studies (Sun Z, Hong Q, Liu Y, et al. Characterization of circular RNA profiles of oviduct reveal the potential mechanism in prolificacy trait of goat in the estrus cycle. Front Physiol. 2022;13:990691. Published 2022 Sep 15. doi:10.3389/fphys.2022.990691). We have supplemented this section in the manuscript(Lines 244-248, 277-279), and the results of RT-RCR experiments were included in the Supplementary Information (Figure S1 Sanger sequencing results of the RT-RCR products of circRNAs).

Point 3: Why not do more research on these differential-expressed RNAs? More wet-lab would be better.

Response: Thank you for your suggestion. Our lab has both wet and dry experiment groups, and the key components screened by the dry experiment group were verified and further studied by the wet experiment team. These two parts are promoted by their respective groups, which improves the efficiency of the research. This article mainly presents the results from the perspective of data analysis, which have been adopted and advanced by the wet experiment group, and will be presented in subsequent articles.

Thank you for your valuable comments. We have made every effort to enhance the manuscript and incorporated some revisions accordingly while ensuring that the content and structure of the paper remain unchanged. We sincerely appreciate the diligent work of yours and hope that our amendments will meet with your approval. Once again, thank you very much for your insightful suggestions.

Reviewer 3 Report

The authors have attempted to study circRNAs involved in the reproductive process in sheep.

Major issues.

The simple summary includes scientific terminology, which will not understood by lay people. Thus, it must be rephrased significantly.

 Minor issues

Please include a new paragraph providing some basic information about the Small Tail Han sheep breed.

2.2. How did you confirm that animals carried the specific genotype?

Figure 3. Please use colours in the graph.

Figure 5. Some of the axis titles are difficult to read.

Please add a new sub-section in the discussion to describe the clinical significance of the findings.

Author Response

Dear Reviewer:

Thank you for your comments concerning our manuscript entitled“Comparative Transcriptomics Identify Key Pituitary Circular RNAs that Participate in Sheep (Ovis aries) Reproduction“ (ID: animals-2528678). These comments are very valuable and helpful for revising and improving our paper, as well as providing important guidance to our research work. We have studied the comments carefully and have made corrections which we hope to meet with approval. Revised portions are marked in red in the manuscript. The main corrections in the paper and the response to your comments are as flowing:

Point 1: The simple summary includes scientific terminology, which will not understood by lay people. Thus, it must be rephrased significantly.

Response: Thank you for your valuable and thoughtful comments. We carefully reviewed the content of the simple summary section, and the most likely confusing part may be the description of FecB++. Small Tail Han sheep of FecB++ genotype still have high fecundity individuals, which is the starting point for us to carry out this research. So we rewrote the simple summary section and gave a brief introduction to the FecB gene. At the same time, we simplified the research methods and conclusions appropriately. We hope these changes ease the difficulty of understanding(Lines 18-28).

Point 2: Please include a new paragraph providing some basic information about the Small Tail Han sheep breed.

Response: Thank you for your suggestion. Based on your suggestion, we moved the brief introduction about Small Tail Han sheep in the first paragraph of the original manuscript to a new second paragraph. On this basis, we added a more detailed introduction about Small Tail Han sheep, including the genotype frequency of FecB in this breed, the average number of lambs and so on. It is hoped that these new additions will give readers a better understanding of the breed(Lines 72-83).

Point 3: How did you confirm that animals carried the specific genotype?

Response: Thank you for your comments. In a previous study by our group, the method for animal and genotyping of the FecB gene by the TaqMan assay was detailed(Wang X, Guo X, He X, et al. Effects of FecB Mutation on Estrus, Ovulation, and Endocrine Characteristics in Small Tail Han Sheep. Front Vet Sci. 2021;8:709737. Published 2021 Nov 22. doi:10.3389/fvets.2021.709737). In our manuscript, the population of FecB++ genotype was selected by this method from a total of 890 Small Tail Han sheep, which also came from our previous study. To reduce confusion for readers, we have added literature citations in the corresponding places(Line 132).

Point 4: Figure 3. Please use colours in the graph.

Response: Thank you for your suggestion. This graph has been modified. We have used more differentiated colors for easier reading(Line 286).

Point 5: Figure 5. Some of the axis titles are difficult to read.

Response: We are sorry for our mistakes in the manuscript. We have rearranged the layout of the content and made the font larger in hopes of resolving this issue(Line 317).

Point 6: Please add a new sub-section in the discussion to describe the clinical significance of the findings.

Response: Thank you for your valuable and thoughtful comments. Following your suggestion, we have added a subsection at the end of the Discussion section presenting the significance of these findings(Lines 493-500).

Thank you for your valuable comments. We have made every effort to enhance the manuscript and incorporated some revisions accordingly while ensuring that the content and structure of the paper remain unchanged. We sincerely appreciate the diligent work of yours and hope that our amendments will meet with your approval. Once again, thank you very much for your insightful suggestions.

Round 2

Reviewer 3 Report

The authors have made significant improvements in the manuscript, by addressing all the issues and by making relevant changes as requested.

The manuscript can be accepted.